Giant ants and their shape: revealing relationships in the genus Titanomyrma with geometric morphometrics

Katzke Julian 1 jkatzke@uni-bonn.de
http://orcid.org/0000-0001-6277-320X Barden Phillip 2
Dehon Manuel 3
http://orcid.org/0000-0001-8880-1838 Michez Denis 3
http://orcid.org/0000-0003-1592-0988 Wappler Torsten 1 4
1 Steinmann Institut für Geologie, Mineralogie und Paläontologie, Rheinische Friedrich-Wilhelms Universität Bonn , Bonn , Germany
2 Department of Biological Sciences, New Jersey Institute of Technology , Newark, NJ , USA
3 Laboratory of Zoology, Research Institute of Biosciences, Université de Mons-Hainaut , Mons , Belgium
4 Naturgeschichte, Hessisches Landesmuseum Darmstadt , Darmstadt , Germany
Mikheyev Alexander
Electronic publication date: 2018 Jan 16
Publication date: 2018
Volume: 6
Electronic Location ID: e4242
Received 2017 Sep 15; Accepted 2017 Dec 18
Copyright: © 2018 Katzke et al.
Copyright year: 2018
Copyright holder: Katzke et al.
License: This is an open access article distributed under the terms of the Creative Commons Attribution License, which permits unrestricted use, distribution, reproduction and adaptation in any medium and for any purpose provided that it is properly attributed. For attribution, the original author(s), title, publication source (PeerJ) and either DOI or URL of the article must be cited.
License URL: https://creativecommons.org/licenses/by/4.0/

Keywords: Palaeontology, Palaeoentomology, Formicium, Formiciinae, Messel, Eckfeld, Wing venation, Formicidae, Fossil ants

Funding: Deutsche Forschungsgemeinschaft (DFG) WA 1492/8-1,11-1 This study was financially supported by the Deutsche Forschungsgemeinschaft (DFG, grant no. WA 1492/8-1,11-1). The funders had no role in study design, data collection and analysis, decision to publish, or preparation of the manuscript.

==============================
Shape is a natural phenomenon inherent to many different lifeforms. A modern technique to analyse shape is geometric morphometrics (GM), which offers a whole range of methods concerning the pure shape of an object. The results from these methods have provided new insights into biological problems and have become especially useful in the fields of entomology and palaeontology. Despite the conspicuous successes in other hymenopteran groups, GM analysis of wings and fossil wings of Formicidae has been neglected. Here we tested if landmarks defining the wing shape of fossil ants that belong to the genus Titanomyrma are reliable and if this technique is able to expose relationships among different groups of the largest Hymenoptera that ever lived. This study comprises 402 wings from 362 ants that were analysed and assigned with the GM methods linear discriminant function analysis, principal component analysis, canonical variate analysis, and regression. The giant ant genus Titanomyrma and the parataxon Formicium have different representatives that are all very similar but these modern methods were able to distinguish giant ant types even to the level of the sex. Thirty-five giant ant specimens from the Eckfeld Maar were significantly differentiable from a collection of Messel specimens that consisted of 187 Titanomyrma gigantea females and 42 T. gigantea males, and from 74 Titanomyrma simillima females and 21 T. simillima males. Out of the 324 Messel ants, 127 are newly assigned to a species and 223 giant ants are newly assigned to sex with GM analysis. All specimens from Messel fit to the two species. Moreover, shape affinities of these groups and the species Formicium brodiei, Formicium mirabile, and Formicium berryi, which are known only from wings, were investigated. T. gigantea stands out with a possible female relative in one of the Eckfeld specimens whereas the other groups show similar shape patterns that are possibly plesiomorphic. Formicidae are one of the most dominant taxa in the animal kingdom and new methods can aid in investigating their diversity in the present and in deep time. GM of the ant wing delivers significant results and this core of methods is able to enhance the toolset we have now to analyse the complex biology of the ants. It can prove as especially useful in the future when incorporated into better understanding aspects of evolutionary patterns and ant palaeontology.

Introduction

Geometric morphometrics (GM) is a recent core of methods aiming at quantifying and analysing the overall shape of a structure. By removing all non-shape variables (i.e. translation, rotation, and scale) and by separately analysing the size and the shape components of the form, the discrimination capabilities of GM are significantly superior to traditional morphometrics because they consider the shape as a whole rather than a collection of independent variables (Adams, Rohlf & Slice, 2004; Slice, 2007; Zelditch, Swiderski & Sheets, 2012; Klingenberg, 2016). The different techniques within the range of GM provide a powerful tool in anatomy, evolutionary biology, systematics, and palaeontology. One essential advantage of this technique is that it is applicable to most zoological taxa (both vertebrates and invertebrates) as well as to most botanical taxa (Viscosi & Cardini, 2011; Bai et al., 2012; Maiorino et al., 2015; Lallensack, van Heteren & Wings, 2016). For example, in bee systematics, morphology and morphometry of the wing usefully discriminate taxa at different levels: specimens, populations, subspecies, species, and tribes (Aytekin et al., 2007; Michez et al., 2009; De Meulemeester et al., 2012; Bonatti et al., 2014; Dehon et al., 2014, 2017). It has even been applied as an inexpensive alternative to molecular analysis to address genetic problems in bee tribes with morphology-derived results being similar to or more detailed than those obtained with mitochondrial DNA (Bonatti et al., 2014). As the method is rooted in morphology, it is especially interesting in palaeontology and palaeoecology where molecular approaches are inapplicable and taphonomy often destroys traditional morphological characters (Gunz et al., 2004; Wappler et al., 2012; Maiorino et al., 2013; Bonatti et al., 2014).

Ants are one of the most dominant groups in the terrestrial animal kingdom, both in diversity and biomass (Hölldobler & Wilson, 1990). The origin of the Formicidae is estimated by molecular clock dating to lie within the Late Jurassic or Early Cretaceous (Brady et al., 2006; Moreau et al., 2006; Ward, 2007, 2014; Moreau & Bell, 2013; Ward et al., 2015; Barden, 2017; Peters et al., 2017). The fossil record of the ants and close relatives starts in the Cretaceous with some early members of modern ant lineages but also an astonishing variety of specialised stem-Formicidae despite constituting only ∼1% or less of all insect fossils (Dlussky, 1999; Barden & Grimaldi, 2013, 2014, 2016; Perrichot, Wang & Engel, 2016). Ant diversity rose quickly in the Paleogene and by the Middle Eocene, most of the extant subfamilies are present in the fossil record (Grimaldi & Agosti, 2000; Dlussky & Rasnitsyn, 2002; LaPolla, Dlussky & Perrichot, 2013; Ward, 2014). The abundance of the Formicidae in the fossil record steadily increases with ants making up to 13% of all insects in Eocene deposits, and over 24% in younger Miocene deposits (Grimaldi & Agosti, 2000; Dlussky & Rasnitsyn, 2002; LaPolla, Dlussky & Perrichot, 2013).

One of the most remarkable formicid attributes is their eusocial behaviour. The scale of what is considered an ant colony ranges from merely around a dozen individuals in rainforests to the formicine ant Formica yessensis (W.M. Wheeler, 1913), which builds up polygynous “supercolonies” that comprise over 300 million adults and over 2.5 million queens alone (Wilson, 1959; Higashi & Yamauchi, 1979; Hölldobler & Wilson, 1990). Their origin and earliest evolution are still unresolved but formicid ecological dominance and abundance in fossil deposits quickly rose due to eusociality and the expansion of angiosperm forests (Moreau et al., 2006; Perrichot et al., 2007; LaPolla, Dlussky & Perrichot, 2013; Rust & Wappler, 2016; Barden, 2017).

Among ant castes, workers, which are much richer in individuals than queens or males, are primarily utilised to discuss biological problems. For Formicidae in general, the worker caste is extensively found in museum collections, they are often found in Cenozoic amber deposits, and they can be collected throughout the year in the field. However, only the queens and males possess wings, which are sometimes used in ant taxonomy to describe those castes (Yoshimura & Fisher, 2007, 2012). Although wings are acknowledged to be quite informative, especially in evolutionary relationships, they are neglected due to the workers’ aptery and only few studies concern wing venation patterns (Brown & Nutting, 1949; Perfilieva, 2000, 2010, 2015; Klingenberg & Dietz, 2004).

At the very core of this work stand the shape of the wing and the venational structures of the giant ants within the extinct formicoid subfamily Formiciinae. The Eocene giant ants are by far the largest known fossil or extant hymenopterans with up to 14 cm wing span and 7 cm body size but their position in the ant tree of life remains unclear (Lutz, 1986; Grimaldi, Agosti & Carpenter, 1997; Archibald et al., 2011). Lutz (1986) put the Formiciinae as the sister group to the Formicinae due to form of the single petiolus and a reduced sting apparatus but phylogenetic reconstructions including giant ants by Baroni Urbani, Bolton & Ward (1992) and Grimaldi, Agosti & Carpenter (1997) placed the subfamily ambiguously due to too many missing characters for the Eocene compression fossils. The Formiciinae currently comprise the genus Titanomyrma and the collective group Formicium, both with three species.

Here, the focus is set on the two described species Titanomyrma gigantea (H. Lutz, 1986) and Titanomyrma simillima (H. Lutz, 1986) from the Messel formation and on undetermined specimens from the Eckfeld Maar for whom affinities towards T. gigantea and T. simillima have been stated, but no thorough description was provided (Wappler, 2003). The recent addition of a single specimen of Titanomyrma lubei S.B. Archibald et al., 2011 from the Green River formation is excluded from the analyses as unfortunately no wing venation is preserved.

Also included are single isolated wings of Formicium berryi (F.M. Carpenter, 1929) and Formicium brodiei J.O. Westwood, 1854 from the Claiborne and Bracklesham Groups, respectively (Westwood, 1854; Carpenter, 1929). The last species in the Formiciinae is Formicium mirabile (T.D. Cockerell, 1920), which was originally thought to be a sawfly and is, like F. brodiei, from the Bracklesham Group (Cockerell, 1920; Lutz, 1990).

The Formiciinae are found exclusively from the latest Ypresian (early Eocene) to the Lutetian (early middle Eocene) of Central Europe (Bracklesham Group, England and Messel and Eckfeld, Germany) and mid-continental North America (Green River Beds, Wyoming and the Claiborne Group, Tennessee). The ages of the oil shales from Eckfeld and Messel are now well established by means of numerical dating: Messel lake is 48.7 ± 0.2 Ma old with a duration of about 640 Ka (Mertz & Renne, 2005) and the Eckfeld maar eruption has an age of 44.3 ± 0.4 Ma (Mertz et al., 2000; Lutz et al., 2010).

Until recently, each giant Eocene ant was attributed to the genus Formicium. For the species from Messel, T. gigantea and T. simillima, the genus “Formicium” as described by Lutz (1986) is no longer valid because Formicium has since been defined as a parataxon that collects species described from wings only (Archibald et al., 2011). Titanomyrma now serves as the orthotaxon for species described from complete or rather complete bodies, most with wings preserved. When described in the future, Eckfeld specimens will also be attributed to Titanomyrma. However, it has to be stated that the revision by Archibald et al. (2011) still leaves nomenclatural problems. The species T. gigantea and T. simillima have been used with the neuter suffix–um. Here and in other appearances, the proper feminine suffix is used. The subfamily Formiciinae also becomes problematic with the treatment of Formicium as a parataxon because the subfamily is now represented by a parataxon as its type species where it should refer to the orthotaxon. Either a new subfamily name is needed or, with more information provided by new studies, giant ants can be incorporated into an existing subfamily. When speaking collectively of the giant ants, the terms Formiciinae and Titanomyrma are used. Formicium is only used when exclusively addressing the wing species.

To investigate if the ant wing is a strong taxonomic character, modern techniques need to be applied to evaluate wing venation as a window into evolutionary pathways and species-level diversity. GM of the hymenopteran wing in general is a thriving method to analyse and discuss morphological issues within and across taxa, both in extant and fossil lineages (Michez et al., 2009; Francoy et al., 2011; Wappler et al., 2012; Bonatti et al., 2014; Dehon et al., 2014, 2017; Perrard, Lopez-Osorio & Carpenter, 2016). However, GM is frequently unused in the Formicidae aside from investigations on cryptic diversity using the ant-body (Csosz et al., 2014; Seifert, Yazdi & Schultz, 2014). The aims of this work are to validate the wing venation of the Formiciinae as a statistically robust set of characters and to distinguish different groups of formiciine ants from the level of species up to the distinction of sex with GM. The data from Messel is expected to show a clear division into the species T. gigantea and T. simillima and undetermined specimens should be assignable to either these species or cluster within a new group. Wing venation for males and females of T. gigantea should differ from representatives of T. simillima, with at least the four described morphogroups that are distinguishable in analyses. These morphogroups are expected to reflect currently defined species diagnoses sensu Lutz (1986). Ants from Eckfeld, which are formerly uncategorised, as well as F. brodiei, F. mirabile, and F. berryi are expected to show affinities to other giant ants and those affinities will be described and discussed thoroughly.

Materials and Methods

Sampling of specimens

Included in this work are wings of a greater collection of fossils belonging to the giant-ant genus Titanomyrma. The detailed analysis of Titanomyrma specimens from Messel and Eckfeld was performed using 399 wings with the addition of wing drawings of the holotypes of F. brodiei and F. berryi provided by Lutz (1986) and F. mirabile provided by Lutz (1990). Due to 40 specimens having both forewings preserved, this study represents 362 ants. A detailed list of specimens with both wings preserved can be found in Tables S1 and S2. The fossils used are taken from the collections of the Hessian State Museum (HLMD), Natural History Museum Mainz (PE = Paleogene Eckfeld, LS = State Collection), and Senckenberg Research Institute and Natural History Museum (MeI = Messel Inventory). Photographs had been taken by Uta Kiel, Sonja Wedmann, and Torsten Wappler and were used as the foundation for assessing landmarks.

Assignments of Titanomyrma ants were done by Lutz (1986) and some were provided by T. Wappler (2016, personal communication) based on wing venation characters and measurements and calculations of the crowding factor sensu Lutz (1986). These specimens were re-evaluated in the GM analysis in combination with Lutz’s (1986) diagnosis of the species (Table S3). A list of the specimens with prior assignment is found in Tables S1 and S3.

For the Messel fossils, 240 out of the 358 wings (219 out of 324 specimens) had been previously assigned to a respective species sensu Lutz (1986). T. gigantea was represented with 180 wings (164 ants) and T. simillima with 60 wings (55 specimens). Only 125 Messel wings had been assigned to either female or male (114 specimens). Females were represented with 109 wings (100 specimens), males are represented with 16 wings (14 specimens).

In a more detailed view, prior to this study, a total of 117 Titanomyrma wings (106 specimens) from Messel had already been assigned to both species and sex. A separation following the classifiers sex and species yielded a total of 87 determined female T. gigantea wings (80 specimens), 17 determined female T. simillima wings (15 specimens), six determined male T. gigantea wings (five specimens), and seven determined male T. simillima wings (six specimens). Specimens from Eckfeld had no prior species assignment, out of the 41 wings (35 specimens), 13 (11 specimens) were assigned as male wings and one wing (one specimen) was assigned to female.

Wing venation and landmark definition

Nomenclature for the wing venation of ants was established by Brown & Nutting (1949) for Formicidae following Ross’s work (1936) that tried to homologise hymenopteran wing venation and erected the terminology. The wing venation terminology used here primarily follows these works with the following modifications: Consecutive numbering is only used for the median and radial-sector veins, the branches of the cubitus are labelled Cua and Cub, radio-medial cross-vein r-m becomes rs-m as it connects the radial sector with the median.

Titanomyrma has a very basic set of veins and almost all wing cells that would occur in a basal representative of the Formicidae are present (Fig. 1A). The only modification to the basal condition in Titanomyrma is the absence of the first radial cross-vein 1r. That vein is also reduced in other ants of the formicoid clade except for some of the Dorylinae, for example Cheliomyrmex (Brown & Nutting, 1949; Bolton, 2016). The position of m-cu in Titanomyrma is highly variable in front or behind the branching of the radial-sector vein. This makes the classical nomenclature of the median veins more difficult as m-cu separates M2 and M3. If a coherent homologisation were desired, it is suggested to use the term Rs + M for any part where the radial-sector and median vein is fused so in different Titanomyrma species the median vein M2 may or may not be reduced.

Figure 1 Titanomyrma wing venation and landmarks.

(A) Schematic drawing with venation nomenclature and cells considered in this study. The wing venation refers to specimen MeI1537, a female T. simillima. (B) The 12 landmarks used in this study digitalised onto specimen MeI10793 (Photo credit: Uta Kiel).

Many compression fossils lack a proper preservation of the wing margins, which is why only the central wing cells 1-2R, 1Rs, and 1M and vein cu-a have been considered for landmark assignment. For each fossil of the Titanomyrma species, 12 landmarks were identified and placed (Fig. 1B). LM1, 2, 3, 4, 7, and 9 surround the radial cells 1R and 2R, which are fused in Titanomyrma to 1-2R due to the reduction of the first radial cross-vein 1r. LM4–8 define the shape of the first radial-sector cell 1Rs. LM8 co-defines the shape of 1Rs as it marks the separation between vein M2 and M3. However, due to the variable position of m-cu, LM8 may be excluded from this observation. The first medial cell 1M is surrounded by LM8, 9, 10, and 12. The position of cu-a is defined by LM11.

Data preparation

Landmarks were digitalised by J. Katzke onto the wing pictures using tpsDig2, version 2.28 (Rohlf, 2016a, Data S1–S2) and analysed using MorphoJ, version 1.06 (Klingenberg, 2011, Data S3–S4). To make all available specimens comparable using GM, all images have to resemble a dorsal view of the right forewing, which is practical for wings as they are almost two-dimensional structures. Wings and imprints appearing right-handed did not have to be altered whereas wings and imprints appearing left-handed were converted by mirroring them on the vertical axis.

Although the Titanomyrma fossils are expected to differ significantly in shape, they also differ greatly in size. T. gigantea and T. simillima females are described to be the real giants with the length of a single forewing measuring 4–6 cm (Lutz, 1986). Male wings are significantly smaller with 2–3 cm (Lutz, 1986). A way to include the size of a specimen along with the shape information is the clear definition of the “centroid size” in a set of landmarks and to make that available for further analyses (Klingenberg, 2011; Zelditch, Swiderski & Sheets, 2012). In this case, a total of 331 wings provided information that could be used to include scaling. A list of these specimens is found in Tables S1 and S2.

To compare the Formiciinae with each other and to assign species to previously unidentified specimens, different classifiers were created using prior identifications and descriptions of the fossils. The specimens are grouped by: locality: Messel, Eckfeld, England, or Tennessee; species: T. giganteum, T. simillima, F. brodiei, F. berryi, or T. sp.; sex: male, female, or undetermined.

Estimating missing data using R

Out of the 402 wings, 80 have missing landmarks, which would exclude them from any further morphometric analysis because applications for data analysis always require the full set of landmarks to calculate the shape differences. Both MorphoJ and programs in “R” are unable to ignore missing landmarks because the analyses focus on the shape as a whole. In paleontological datasets, missing data is a commonly encountered problem and several computational methods have been made available and applied successfully over the last 13 years (Gunz et al., 2004; Maiorino et al., 2013, 2015; Hopkins & Pearson, 2016). A list of specimens with missing data can be found in Tables S1 and S2 and the missing landmarks themselves are listed in Data S1.

Missing landmarks were estimated using R, version 3.3.1 (R Core Team, 2016) package “Morpho,” version 2.5.1 with the command “fixLMtps” (Schlager, 2017). The command uses thin-plate-spline-interpolation techniques according to the inverted Procrustes distances between landmark observations of, in this case five, most similarly shaped individuals (Schlager, 2017). The .tps-file created with tpsDig2 is read by applying a variation of the Morpho program “readallTPS,” which was manipulated to read the “IMAGE=”-lines in the file for identification of the specimens instead of the “ID=”-lines (Data S5).

Three wings from Eckfeld and 77 wings from Messel that are included in the dataset have missing landmarks. The estimations for the Eckfeld specimens were performed separately from the Messel specimens and subsequently the files were appended again using tpsUtil, version 1.74 (Rohlf, 2017). As the estimates only have the separation of locality, they could relate to the shapes of other species in the genus Titanomyrma if there was not enough shape difference between species or if there were undiscovered species. The estimated landmarks are collected with the manually placed ones as the final dataset in Data S2.

Data analysis

Before performing different analyses, the curved shape space, which is defined by the raw data, was transformed into the Euclidean distances tangent space via full Procrustes superimposition, which is the crucial step in GM analysis (Kendall, 1977; Bookstein, 1997; Rohlf, 1999). Principal axes align the data during the performed Procrustes fit (Rohlf, 1999). It is theoretically possible that the variation in the dataset is too large for the tangent space being approximate to the curved shape space. By calculating the regression slope and the correlation coefficient between the Procrustes distances in the shape space and the Euclidean distances in the tangent space, it is possible to ascertain whether or not the variation amplitude in the dataset is small enough to perform further analyses (Rohlf, 2015). This analysis was performed with the software tpsSmall, version 1.33 (Rohlf, 2016b).

The dataset yielded from Data S2 was divided into subdatasets (SDs 1–12) in MorphoJ by implementing different classifier information (Tables S1 and S2). The 80 wings from specimens with both wings preserved were treated as individuals in most of the analyses. In order to validate this treatment, variation in a subsample of 17 T. gigantea previously determined females with preservation of both forewings was analysed (SDs 1 and 2; Data S3; Table S1). For a shape-related analysis of the species T. gigantea and T. simillima, Messel specimens were collected in SDs 3–8 (Data S4). Questions concerning assignment of species and sex with GM and sexual dimorphism were investigated. Shape affinities and size of Titanomyrma groups were analysed using SDs 9–12 (Data S4).

Different methods of GM were applied to the subdatasets to gain insights into the relationships between giant ants and to test existing classifications (Table 1). The methods applied here are well established in works using GM to analyse wing venation patterns and wing shape in Hymenoptera (Perfilieva, 2010; Bonatti et al., 2014; Dehon et al., 2014, 2017).

Table 1 Subdatasets used for the GM analysis.

Subdata-set	Number of wings contained	Types of specimens contained	Classifiers used	Analyses performed	For investigating	
SD 1	34	T. gigantea females	Predetermined	PCA, CVA (specimen, l/r wing)	Variability within specimens	
SD 2	46	T. gigantea females, T. gigantea males, T. simillima females	Predetermined	LDA	Variability within specimens	
SD 3	358	All Messel specimens	Predetermined	PCA	Shape discrimination between Messel species	
SD 4	257	All T. gigantea	After species assigned	PCA	Shape discrimination within T. gigantea	
SD 5	101	All T. simillima	After species assigned	PCA	Shape discrimination within T. simillima	
SD 6	290	Messel specimens with scale	After full assignment	Regression, then: PCA, LDA	Sexual dimorphism in Messel species	
SD 7	257	All T. gigantea	After full assignment	PCA	Shape discrimination within T. gigantea	
SD 8	101	All T. simillima	After full assignment	PCA	Shape discrimination within T. simillima	
SD 9	402	All specimens	After full assignment	PCA, LDA	Shape discrimination between all specimens	
SD 10	331	All with scale	After full assignment	PCA, Regression, LDA	Shape and size discrimination between all specimens	
SD 11	141	All T. simillima, Eckfeld males, PE-1998-17	After full assignment	PCA, CVA	Shape affinities within “simillima-morphogroup”	
SD 12	41	All Eckfeld specimens	After full assignment	PCA, CVA, Regression, LDA	Shape affinities within the Eckfeld specimens	
Note:

Subdatasets (SDs) were all created within MorphoJ out of the overall dataset consisting of 402 wings. Procrustes fits were performed for each SD. The only analyses performed with data generated from another analysis were done in SD 6 and SD 10 after regression to investigate the sexual dimorphism in Titanomyrma ants.

Principal component analysis and canonical variate analysis

Principal component analysis (PCA) was widely used in this work to visualise and investigate variation in the dataset. A PCA transforms the total possible observations and reduces them to a data dependent number of “principal components” (PCs) that explain the total variation within a dataset (Zelditch, Swiderski & Sheets, 2012). In contrast to PCA, the canonical variate analysis or CVA calculates and visualises the differences between a priori groups (Klingenberg, 2011; Zelditch, Swiderski & Sheets, 2012). PCA visualises variation in the dataset and CVA visualises differences between groups (Zelditch, Swiderski & Sheets, 2012).

Linear discriminant function analysis

The linear discriminant function analysis (LDA) is a method of multivariate analysis of variance. LDA uses the mean shapes of a priori defined groups to make an assertion of the significance of the groups (Klingenberg, 2011). The LDAs were performed using a cross-validation approach within MorphoJ (SDs 2, 6, 9, and 12) and results were collected within Table S4.

Each separable group, assumed after classifier criteria, was tested against the other groups to estimate shape-related associations among the groups. Eckfeld specimens were tested as their own group. The effectiveness of the cross-validation assignment of the groups is measured by the hit-ratio (HR) of how many specimens could be reassigned to their original group.

Regression modelling to investigate size-related effects

To test whether there is a significant influence of size on shape that can distort the differentiation of species, sex, or both in combination, statistical regression can be applied in MorphoJ to analyse effects of allometry, the relation between size and morphology (Klingenberg, 2011). The aim of the regression is to analyse relationships between dependent and independent variables within a dataset. In this case, shape is the dependent variable linked to the Euclidean distances gained from Procrustes superimposition. Centroid size is at least theoretically independent from shape but shape can be predicted for any centroid size if there is allometry (Klingenberg, 2011). A residual shape, which is the deviation from the prediction, remains. The residual part in shape does not covary with the centroid size or actual size (Klingenberg, 2011). When differentiable groups are present within the dataset, it is possible to perform a pooled within group-regression to see whether or not their different sizes are affecting the shape differences among the groups (Klingenberg, 2016). This was done with SD 6, 10, and 12 to investigate sexual dimorphism in Titanomyrma. The regression as a method to test size influence in a dataset is only advisable if there actually is an association between increasing size and shape change (Zelditch, Swiderski & Sheets, 2012; Klingenberg, 2016).

Results

Using the Procrustes fitted data in the analysis is possible as the Euclidean distances in tangent space approximate the Procrustes distances in shape space for a total of 402 Titanomyrma/Formicium wings. This is indicated by the regression slope being very close to 1 (0.9967) and an equally high correlation coefficient of 0.9999.

Variation within single specimens

At first, the smaller subsample of 17 determined T. gigantea female ants was analysed, of which all 17 show both wings preserved (SD 1). In a PCA, PCs1–6 describe more than 5% variance each and PCs1–3 describe more than 10% variance each. The main variation does not represent a separation in left and right wings. It also does not depict single ants being severely different from the others. The highest variation described by PC1 (31.25%) comes from the relative size of wing cell 1Rs. The 17 T. gigantea females show a very homogenous shape as indicated by the shape changes of the PCs. In the PCs, there is no separation of distinctive wing pairs at all. A clear separation of a single ant, MeI409, is the result of a CVA of the subsample grouped after specimen (CV1 = 86.99%). The CVA results in 16 CVs (17 groups) that are able to differentiate the groups but with 87% of the results being insignificant (p > 0.05). A separation of left and right wings in a CVA is not possible. Using the LDA in SD 2, the permutation tests result in insignificant values with only one significant result (p < 0.05).The allocation of wing pairs to each other in a cross-validation approach is unsuccessful throughout tests regarding each wing pair against other ant wings.

The independence of shape between left and right wings of single specimens creates a confining factor for GM as a quantitative method. It is not possible to assign an isolated Titanomyrma wing to its counterpart. Cutting the specimens that are isolated wings in half would be a means to estimate the least amount of ants that are preserved. However, during biostratinomy a once-connected pair of fragile ant wings is influenced by several confounding factors such as predators, currents, or different sinking speeds. From a taphonomical point of view, it is highly doubtful that any of the isolated wings preserved has a matching counterpart in the same dataset.

Shape patterns within the genus Titanomyrma

Shape discrimination between Messel species

The described species with specimens predetermined as T. gigantea and T. simillima are clearly separable by comparing their wing shape in a PCA (SD 3, Fig. 2A). Moreover, all undetermined specimens from Messel cluster within the group of either T. gigantea or T. simillima (Fig. 2A). Out of the predetermined specimens, nine were reassigned to the other species (Table S3). Wing shape of the undetermined specimens can easily assign them to the two species as they cluster within the range of either T. gigantea or T. simillima. PCs1–3 each explain more than 5% of the total variance in the dataset. PC1 represents 58.31% variance and PC2 represents 13.60% variance, and they both describe clear shape trends in distinguishable groups. There is a separation of larger and smaller specimens in PC2 (Fig. 2B). The two species T. gigantea and T. simillima are separated in PC1, which describes differences in shape that are clearly observable in the mean shapes of the two species (Fig. 2A). The most important vein for the distinction between the species is m-cu, which is virtually in line with Rs1 in T. gigantea similar to Camponotus ants. A small distance of m-cu occurs in some individuals, especially smaller specimens. Another difference is the position of cu-a, the connection between the cubital and anal veins that is also used as a delimitating factor in other ant groups (Brown & Nutting, 1949; Perfilieva, 2010, 2015). In T. simillima, cu-a is somewhat more variable and nearer towards the ant-body than in T. gigantea where most often cu-a and M1 build up a junction with M + Cu and Cu. The vein rs-m, which connects the radial-sector and the median veins is well expressed in T. simillima and reduced in T. gigantea so that those veins are directly in contact and appear in coalescence. In general, T. gigantea is denser in its wing venation than T. simillima, which corresponds to the higher crowding calculated by Lutz (1986).

Figure 2 PCA of Titanomyrma specimens from Messel.

The analysis was performed in SD 3. (A) Wireframes represent the mean shapes of the respective species. High values in PC1 (58.31%), where T. simillima clusters, signify the branching of m-cu—farer off from M1, more distance between M1 and cu-a, a more pronounced rs-m, and larger relative sizes of cells 1-2R and 1Rs. Low values in PC1, where T. gigantea clusters, signify the opposite. Problematic specimens are marked with crosses (see Table S3). (B) High values in PC2 (13.60%) mean a narrower wing in general, expressed by the shape shift of LM1, 2, 3, and 12. The lollipop graph dots indicate the mean shape of Titanomyrma specimens from Messel, lines stretch out in positive PC-value-shape change and end in with the value of 0.10 in PC2.

MeI4060 is the only intermediate specimen in a PCA (Fig. 2A). The wing is well preserved but shows white patches in all the critical veins, which made the digitalisation of landmarks more difficult. An offset of the veins Rs1 and m-cu puts the individual into the species T. simillima and additionally, cu-a is also a bit distant from M1.

As no third larger group was detected in the Messel specimens, out of 127 undetermined wings, 81 were assigned to T. gigantea (n = 257) and 46 to T. simillima (n = 101), which is statistically robust using the LDA in SD 9 with hit ratios (HRs) of 100% and significant p values (<0.0001) in both Procrustes and Mahalanobis distances (Table S4).

Shape discrimination between sex in Messel specimens

There is sexual size and shape dimorphism in Titanomyrma related to PC2 of the Messel PCA (Fig. 2B; SD 3). The shape difference may be attributed to allometry and the size difference of females and males. For the sexual assignment of the undetermined specimens, when present, size information was used after species determination. Otherwise, PCA for each species could be applied to assign the specimens to either male or female (SDs 4 and 5). Out of the predetermined specimens, two were attributed to the wrong sex (Table S3). Assignment to sex provides 64 new T. simillima females, 16 new T. simillima males, 123 new T. gigantea females, and 37 new T. gigantea males. Regression pooled within both species suggests a linear combination of shape and size and that size difference predicts shape (SD 6). This regression results in 20.54% predicted values and the sex difference is almost entirely contained in the prediction. The residuals leave no sexual shape variation, only that of the species. The shape change, which is realised in increasing centroid size, resembles the shape change in PC2 from the PCA of Titanomyrma specimens from Messel (Fig. 2B; SD 3). LDA regarding sex on the regression residuals is grossly insignificant (p = 0.79) whereas LDA on the prediction assigns 100% of the males correctly and 84.44% of the females (p < 0.0001, SD 6; Table S4). In Messel, Titanomyrma males and females differ in their relative wing width, which affects both species and the males are always much smaller than the females. The shape of the wing venation of the males is generally congruent with the typical condition for the respective females (see Fig. 2A). That T. simillima and T. gigantea share the same pattern of shape dimorphism despite being differentiable in shape, speaks for a size-related allometric origin of that dimorphism.

However, besides the female wings being narrower, there are also other slight shape differences in the sexes for each species that make them distinguishable using GM. In general, the differences between T. simillima males and females are more numerous and easier to observe than the ones between T. gigantea representatives. The alignment of m-cu and Rs1 is not as progressed in T. gigantea males as in the females (Fig. 3A). In T. simillima, M2 is relatively larger in males as well as rs-m. Moreover, the positioning of cu-a is generally more proximal to the wing base in male specimens of T. simillima (Fig. 3B). Despite a trend of narrower female wings, results of PCA after assignment to sex are diffuse for the individual species; in T. gigantea (SD 7), no relation of PCs offers a clear distinction between sexes whereas a distinction in T. simillima (SD 8) is still possible (Figs. 3C and 3D). Strong outliers in these analyses are included in Table S3.

Figure 3 Shape patterns and variation between Titanomyrma females and males.

In Titanomyrma, the sexes of the two described species share a common pattern of sexual shape dimorphism (A) Procrustes fitted mean shapes of T. gigantea females and males in comparison (SD 7). The female shape is narrower than the male shape. (B) Procrustes fitted mean shapes of T. simillima females and males in comparison (SD 8). The female shape is also narrower than the male shape. (C, D) Problematic specimens are marked with crosses (see Table S3). (C) A PCA of all 257 T. gigantea wings (SD 7). The main variation does not well describe a shape separation between T. gigantea females and males. (D) A PCA of all 101 T. simillima wings (SD 8). Shape variation is more distinct for T. simillima females and males but there is a severe outlier.

Following species assignment, the LDA improves upon sex discrimination relative to shape affinities revealed by principal components. In the cross-validation approach of the LDA, the specimens are significantly well separable (SD 9; Table S4). T. simillima females show a 98.25% HR (n = 80) and T. simillima males show a 100% HR (n = 21). T. gigantea females show a 99.03 HR (n = 207) and T. gigantea males show a 98% HR (n = 50). These results suggest that there is in fact solid shape discrimination between males and females of both species and that sexual dimorphism in Titanomyrma affects both size and shape.

Shape trends of all specimens and assignment of Eckfeld specimens

In a second approach performing a PCA on all Titanomyrma specimens available (SD 9), T. gigantea from Messel clusters on one side of the plot and all the other specimens gather on the other side in PC1 (Fig. 4A). One exception constitutes PE_1994_167-LS, which appears within the range of T. gigantea. PE_1994_167-LS has the typical crossing of Rs1 and m-cu with the median vein, which is never observable in T. simillima or the other Eckfeld specimens. The unambiguous distinction of T. gigantea from the other groups is also supported by the LDA, which results in 100% HRs (p < 0.0001, SD 9; Table S4). All the Titanomyrma groups are separable using a combination of shape variables and size (SD 10; Fig. 4B). T. simillima males are the smallest wings with about 23 mm (Lutz, 1986). Most of the Eckfeld specimens have roughly the same size as T. gigantea males with about 27 mm but they differ in shape (Lutz, 1986). F. brodei and F. berryi also exhibit a wing length of about 26–27 mm (Lutz, 1986). Female wings of the species T. gigantea measure about 60 mm, whereas female T. simillima wing only measure about 45 mm (Lutz, 1986). F. mirabile is closer in size to T. gigantea females with about 54 mm but closer in shape towards T. simillima (Fig. 4A; Lutz, 1990). Two specimens from Eckfeld, PE_1994_167-LS and PE_1998_17-LS are much larger than the rest of the Eckfeld specimens and are interpreted as females. In size and shape, PE_1994_167-LS is well within the range of T. gigantea females. PE_1998_17-LS is just above the size of T. simillima females and is close in shape. The other 37 Eckfeld wings were all classified as males because they are similar in size and shape. Five Eckfeld specimens fall in the range of small T. simillima females but are nevertheless assigned as males: PE-1990-582-LS, PE-1992-258-LS, PE-1992-506-LS, PE-2000-15-LS, and PE-2000-18-LS. This could be a misinterpretation but as seen in Messel, the size differences between males and females are drastic, more drastic than it would be in these five specimens. Moreover, no further shape discrimination is detectable between these five larger specimens and the others (Fig. 4A and see Fig. S1; SD 12). The overall determination as Eckfeld males is confidently undertaken not only because of their size ranging within that of T. gigantea males, but also because of specimen PE_2000_3-LS, of which Wappler (2003) thoroughly described a male genital apparatus.

Figure 4 Shape and size variation among Titanomyrma and Formicium specimens.

(A–C) Five larger Eckfeld males are highlighted with crosses. (A) A PCA of all specimens reveals close shape associations between Eckfeld specimens, T. simillima females, F. brodiei, and F. mirabile. F. berryi appears in an intermediate position. The overall pattern resembles the PCA of the Messel specimens (Fig. 2A). The analysis was performed in SD 9. (B, C) The analyses were performed in SD 10. (B) The independent variable Centroid Size is regressed over the dependent shape variable. T. gigantea is distinct in shape, signified by a higher Regression score. Eckfeld males and T. gigantea males have a similar size range. There is also a size overlap between Eckfeld males and T. simillima females. (C) A PCA including all specimens with scaling information, grouped after classifiers locality, species, and sex. This PCA was performed with the residuals of an attempt to correct for size by pooled-within-group regression of shape over centroid size. Most of the variation is still between T. gigantea and the others. Female and male specimens cluster apart from each other in PC 2.

Linear discriminant function analysis is not applicable to groups with only one specimen, which concerns F. brodiei, F. berryi, F. mirabile, and the larger Eckfeld specimens. For all discriminable groups that have more than one specimen (T. gigantea females; n = 207, T. gigantea males; n = 50, T. simillima females; n = 80, T. simillima males; n = 21, Eckfeld males; n = 39), the results of the LDA show combined HRs of more than 97% for each group against the others with significant p values (SD 9; Table S4). In a PCA, the largest specimens cluster with the rest of T. sp. from Eckfeld and F. brodiei in PC2 (Fig. 4A). The shape change realised in PC2 is the same as in Fig. 2B so Eckfeld males have narrower wings than males from Messel and no Eckfeld female wing is explicitly narrower (Fig. 4A). Thus, narrow wings are not necessarily a Titanomyrma trait for the larger females. The five larger Eckfeld males are also scattered across the cluster (Fig. 4A). In a PCA concerning only Eckfeld specimens, Eckfeld females cluster on opposite sides in PC1 with only 20.85% variation, leaving the males slightly intermediate (Fig. S1; SD 12). Apart from that and size, no sexual shape dimorphism is evident in Eckfeld.

An attempt for size-correction by pooled within group-regression slightly alters the shape trends in Titanomyrma specimens from Messel and Eckfeld (SD 10; Fig. 4C). Eckfeld specimens, which are roughly the same size as T. gigantea males or smaller F. simillima females cluster with other males in PC2 after size correction (Fig. 4C). Most shape variation, represented by PC1 in the PCA, separates T. gigantea from any other group of giant ants. This is independent from size. Before allometric correction, PC1 has loadings of 58.44% and after size correction PC1 contains 60.65% of the total variation. Results of the LDA are improved after size correction in the groups that are tested. The HRs are higher with 98.71% and all p values show strong significance (<0.0001). The realised shape change for the males however, is ambiguous: relatively smaller wing cells and a greater distance between cu-a and M1. After the attempt for size correction, there seems to be a separation of male and female Titanomyrma specimens, but these analyses are based on altered data and they do not represent the natural shape of the wings.

Ten different groups of Titanomyrma ants are presented in this study. However, five are only represented with single specimens (Fig. 5). For T. gigantea females and males, for T. simillima females and males, and for T. sp. males from Eckfeld, the similarity of many wings has been quantified by GM analyses. Different aspects of the wing venation and the shape of wing cells 1-2R, 1Rs, and 1M characterise the five larger groups and the five single specimens (Table 2). Especially T. gigantea is distinct by two crossings of veins: 1. M+Cu, Cu, M1, m-cu, and 2. M1, M3, Rs1, m-cu. The reduction of rs-m however, is observable both in T. gigantea and T. sp. but in T. gigantea females it is most advanced. A more generalised pattern of wing venation is observable in giant ants that do not belong to T. gigantea.

Figure 5 Titanomyrma and Formicium groups and individuals and their shapes.

All Procrustes-fitted shapes yielded from SD 9. (A, B, D, E, G) Procrustes fitted mean shapes of Titanomyrma groups represented with more than one specimen. (C, F, H, I, J) Individual Procrustes-fitted shapes of specimens with unique characteristics. PE_1994_167-LS (C) and PE_1998_17 (F) are specimens from Eckfeld that far exceed the size range of the rest of the specimens from Eckfeld (see Fig. 4C). F. berryi (H) is the only known giant ant wing with preserved venation from North America. F. brodiei (I) and F. mirabile (J) are specimens in the size range of a male (26 mm) and a female (54 mm) from Southern England. (H–J) The shapes are based on landmarks digitalised onto interpretative drawings of the fossil specimens and must be taken with caution.

Table 2 Wing venation patterns in Titanomyrma and Formicium groups and individuals.

Group or individual	Crowding	Approximate size of wing (mm)	Alignment of M1, and cu-a	Alignment of Rs1, M1, Rs + M, and m-cu	Reduction of M2	Reduction of rs-m	Wing broad or narrow	
T. gigantea females	++1	601	++	++	/	++	Narrow	
T. gigantea males	+1	271	++	+	/	++	Broad	
T. simillima females	−−1	451	−−	−	++	−	Narrow	
T. simillima males	−1	231	−−	−−	+	−−	Broad	
Eckfeld males	Varies3	27	−	−−	−	+	Narrow	
PE_1998_17-LS	/	>45	++	−−	−−	−	Narrow	
PE_1994_167-LS	/	>60	+	++	/	−	Narrow	
F. brodiei	+1	261	−	−−	−−	−−	Narrow	
F. berryi	−1	261	+	−−	−−	+	Broad	
F. mirabile	−2	542	+	−	+	−	Narrow	
Notes:

The observations are ranked from well expressed (++), over expressed (+) and not expressed (−) to not expressed at all (−−). All observations from ++ to −− are in relation to the other groups and subjective. Crowding is measured by dividing the distance between crossings R + Sc/Rs1 and rs-m/M4 by the total length of the wing. Lower values indicate a more crowded wing. Reduction of M2 is not assessable in T. gigantea and PE_1994_167-LS due to the position of m-cu before the branching of Rs2-3.

1 Lutz, 1986.

2 Lutz, 1990.

3 Wappler, 2003.

Affinities among T. simillima and Eckfeld specimens

There are stronger similarities in shape between T. simillima and the Eckfeld specimens (except for PE_1994_167-LS) than there are with T. gigantea. The main variance that is explained by principal components in all present Titanomyrma groups is always the difference between T. gigantea and others. The similarities and differences in this “simillima-morphogroup” (comprising all non-T. gigantea specimens) have to be investigated without T. gigantea affecting the total variation. Separating the simillima-morphogroup (SD 11) from the T. gigantea-data reveals a clearer pattern in a PCA (Fig. 6A). Eckfeld specimens were determined as males according to Wappler’s (2003) identification of male genitalia but males of T. simillima are clearly separated from the bulk in PC1, which represents the greatest variation in the dataset but with only 26.55% variation. The negative loadings of PC1 that separate males of T. simillima are mainly described by wider wing-cells, a relatively larger rs-m and a greater distance between cu-a and M1 (Fig. 6B). By comparing the mean shapes of the Eckfeld specimens and T. simillima males, the same shape differences are evident (Fig. 6C). However, except for relative size, wing cells 1-2R and 1M are similarly shaped. Aside from the relative length of rs-m, cell 1Rs is very similar in the angle between Rs2-3 and Rs4 and the relative length of these veins and M2, which is almost always unrecognisable in other Titanomyrma groups. The cross-validation approach of the LDA separates T. simillima from Eckfeld males with a 100% HR (p < 0.0001). The distinction of the Eckfeld males from T. simillima males is supported by the LDA with 94.87% HR (p < 0.0001, SD 9; Table S4). However, this is the lowest individual HR in the LDA among all the groups tested.

Figure 6 Shape analysis of the simillima-morphogroup.

All analyses were performed to investigate the shape similarities and differences between Eckfeld specimens (except for PE_1994_167, which is more similar to T. gigantea) and T. simillima. (A, B) Analyses were performed using SD 11. (A) PCA of the simillima-morphogroup separates T. simillima from the others in PC1 (26.55 %). (B) Low values in PC1, which separate T. simillima males, represent a cu-a more proximal to the ant body and relatively larger veins Rs+M and rs-m. (C, D) Analyses were performed using SD 11. Mean shapes of T. simillima males (C) vs. Eckfeld males and mean shapes of T. simillima females (D) vs. Eckfeld males. (E, F) Analyses were performed within SD 11. (E) CVA of the simillima-morphogroup separates the Eckfeld males in in CV1 (55.06 %). A closer association of Eckfeld males with PE_1998_17-LS is indicated by the same loadings in CV1. (F) Differences between Eckfeld specimens and T. simillima are: cu-a and M1 closer together, M2 well expressed, rs-m more reduced.

Principal component analyses reveal that T. sp. and T. simillima females share almost the same shape patterns and that there is little variation between them (Figs. 4A and 6A). Still, in comparison to female T. simillima, males from Eckfeld have a less crowded cell 1Rs, the veins Rs + M and M2 are short but developed, and intriguingly, vein rs-m is shortened (Fig. 6D). This reduction of rs-m and the size of the Eckfeld males are shared with T. gigantea whereas the general shape pattern is close to T. simillima. The LDA is able to separate Eckfeld males from T. simillima females with 97.44% HR and T. simillima females from Eckfeld males with 96.25% HR. In a CVA of the simillima-morphogroup, CV2 (41.36%) collects T. simillima females and Eckfeld males but discriminates T. simillima and T. sp. from Eckfeld (including PE_1998_17-LS) in the positive and negative values (Figs. 6E and 6F). The CV1 shape change is almost identical to the comparison of T. simillima females and Eckfeld males, which is probably due to the higher numbers of individuals. Nevertheless, this represents a difference between T. simillima and the Eckfeld specimens.

Summarised, T. sp. from Eckfeld has a reduced rs-m in the pattern of T. gigantea and M2 is well identifiable in contrast to T. simillima. The relatively narrow wings and the similarity in shape to the single female specimen PE_1998_17-LS leave a sexual shape dimorphism as clearly seen in Messel questionable for T. sp. from Eckfeld. In general, the wing shape is more generalised as it is in T. simillima, F. brodiei, and F. mirabile although the groups are evidently separated from each other in terms of age and locality. Intriguingly, F. brodiei has the same size as the Eckfeld males and F. mirabile has approximately the same size as PE_1998_17-LS.

Distribution of giant ants within localities

All specimens are assigned to species and sex with the Eckfeld specimens being mostly males of probably the same species with two exceptions. Specimens with two wings preserved are represented two times each in the dataset and have to be halved for an individual-level count. Undetermined specimens from Messel are assigned with PCA (SD 3); their sex is determined by size and PCA (SDs 4–8). PE_1994_167-LS from Eckfeld shows affinities towards T. gigantea and its size is suggestive of a female wing (SDs 9 and 10). PE_1998_17-LS is assigned as a female that shows affinities towards T. simillima and is more similar to the 33 males that constitute the rest of the Eckfeld dataset (SDs 9–12). Figure 7 shows a list and a pie chart of the distribution of specimens in the dataset. The ratio of males is about 20% males in Messel. In Eckfeld, the specimens consist of 80–94% males, depending on whether or not the five specimens that have the size of smaller T. simillima females are interpreted as Eckfeld females. In Messel, 29% of all Titanomyrma ants belong to the species T. simillima. Lutz (1986) concluded a similar ratio in Messel but the proportion of T. simillima as a species has decreased with more specimens assigned.

Figure 7 Total and relative abundance of giant ants in different localities.

(A) A count of all ants included in the dataset with two-wing specimens already halved for the count. (B) Relative abundance of different Titanomyrma types in Messel. (C) Relative abundance of different Titanomyrma types in Eckfeld.

Discussion

Landmarks for ants and ant fossils

Using a system of 12 landmarks in a formicid with a generalised condition in its wing venation bears several advantages especially looking at different qualities of preservation in the fossil record. Nevertheless, regarding the evolutionary development of different ant taxa, for a general set of landmarks, other configurations of landmarks are possible. In a generalised ant wing dataset where all critical points in the venation are considered, a set of 23 + 2 landmarks is possible (Fig. 8). The other 11 landmarks, in addition to the 12 landmarks used here, mostly relate to origins and apices of the horizontal veins to encompass the wing shape as a whole in contrast to using only three wing cells and cu-a. Additional landmarks for the complete description of the most ancestral ant wing would include two landmarks for the anterior and posterior ends of 1r, a vein most often reduced but still present even in extant Formicidae like the South American army ant Cheliomyrmex morosus (F. Smith, 1859) or sometimes found as atavisms in other taxa (Brown & Nutting, 1949). Cross vein 1r as an atavism extends to Titanomyrma as seen in the holotype of F. mirabile and the female T. gigantea specimen MeI14311.

Figure 8 Twenty-three landmarks digitalised onto complete wings of T. gigantea specimens.

(A) Specimen MeI12091 is a well-preserved female (Photo credit: Uta Kiel). (B) Specimen MeI3362 is a well-preserved male with only the petioles and gaster missing (Photo credit: Uta Kiel).

Alternative numbers for analysed landmarks are as sparse as publications on ant wing GM. Perfilieva (2010) used a set of 13 landmarks for a study including all major ant subfamilies, and Perfilieva (2015) used a set of 16 landmarks for a study concerning Myrmeciinae and Ponerinae. The lower number in the former publication is due to the fact that many ant lineages reduced cell 1Rs so landmarks concerning this cell are often not placeable. It should be noted that it is not possible to include landmark loss, and corresponding cell loss, into morphometric analysis. The additions in shape assessment compared to our study encompass the overall length of the wing and cell 1Cu, which is shaped by veins Cu, A, and cu-a. Including the overall length of the ant wing by using the apex and the base of the wing for landmarks is helpful in visualizing relative sizes of the wing cells. These landmarks make crowding more visible. However, the higher crowding for T. gigantea estimated by Lutz (1986) is also represented in our study by PC1 in PCAs that include T. gigantea amongst other groups. Additional landmarks are not necessary to assess the relative sizes of wing cells when they are compared with other groups.

The shape of the cubital wing cell 1Cu is not discriminated in this study. A separation between Ponerinae and formicoid ants could be observed by the shape of 1Cu (Perfilieva, 2010). Further including this cell could give interesting insights in general shape trends of the ant wing venation as this cell is highly variable among ant taxa and especially the position of cu-a is notoriously variable even in left and right wings of a single ant (Brown & Nutting, 1949; Wappler, 2003).

In this work, a set of fewer landmarks is acceptable or even desirable because there is a large dataset that consists of only fossil material where in most of the cases the apex of the wing is not preserved and the origin of the wing is usually neither preserved nor identifiable over the ant body. Thus, a reduction in landmark number allows for more specimens to be included. In this dataset, only 25 out of 399 wings are complete enough to assess a full set of 23 landmarks (Table S2). The shape variation in the cells 1-2R, 1Rs, and 1M with 12 landmarks however, is sufficient enough to discriminate between the types of Titanomyrma specimens but for a study among different ant taxa, not all of the 12 landmarks may be assessable due to reduction of wing veins.

Using GM in the fossil record benefits from its versatility but relies on the complete set of landmarks to be applicable. Twenty percent of the wings included could only be included because of missing landmark estimation. Incorporating incomplete specimens not only enlarges sample size, it also improved the results in contrast to removing incomplete specimens in studies with manufactured missing data (Arbour & Brown, 2014). Out of the 36 wings that were problematic in variation analyses (Table S3; from Figs. 2 and 3), 13 have missing values that were estimated. That increased percentage is only based on a visual sample (Figs. 2 and 3) but hints at a loss of distinct venation characters, which especially affects the sexual shape dimorphism in T. gigantea (Fig. 3A).

New insights into Titanomyrma

Lutz (1986) based the distinction between species T. gigantea and T. simillima on size and characters in the wing venation, which can be made quantifiable using GM and the whole spectrum of ants from Messel is covered with these two species. An allometric trend in Messel can be observed that the smaller males have broader wings, which was not previously acknowledged in studies that only considered crowding (Lutz, 1986, 1990; Wappler, 2003). Aside from smaller shape differences and wing width and size, males are characterised by a stout, almost round gaster and a relatively large, pointy head with filiform antennae (Lutz, 1986). Females are larger, have a narrower wing, a more lengthened gaster, and relatively short antennae (Lutz, 1986).

The sex dependent shape differences in T. gigantea are much smaller than the differences between species. One of the most dominant factors in sexual difference is size, which is also heavily supported by regression models. There is a trend that larger ants concentrate their first medial, first radial, and first radial-sector cells in the centre of the wing which is called “crowding” (Lutz, 1986, 1990). A stronger signal of allometry could separate the ants in shape just according to their different sizes. However, Lutz (1986) observed that Titanomyrma males’ crowding values fit to those of the females despite being only about a third in size. The crowding of wing cells 1-2R and 1Rs is species dependent, and so is the overall shape of the wing venation.

When speaking of crowding, ants from Eckfeld showed similar distributions as ants in Messel and by measuring the wings they could be incorporated into existing species (Wappler, 2003). Here, the relative sizes of the wing cells fit exclusively better to T. simillima. Titanomyrma sp. from Eckfeld has its own shape pattern. Except for PE_1994_167-LS and a somewhat reduced rs-m no further affinities to T. gigantea could be ascertained for any of the specimens included here according to GM. Males from Eckfeld do not seem to follow the allometric trend as it is observable for males in Messel although also being considerably smaller than their putative female counterpart, PE_1998_17-LS. However, to say that Titanomyrma sp. from Eckfeld does not exhibit a sexual shape dimorphism is difficult to assess as the female sample size is too small for a thorough analysis.

Not only because of the smaller sizes and their similar shape, there is evidence to say that except for PE_1994_167-LS and PE_1998_17-LS, all the specimens from Eckfeld are males. Surprisingly, the first male genital apparatus of a Titanomyrma ant, PE_2000_3-LS, could be described based on a specimen from Eckfeld distinguished by its large size relative to the abdomen of the ant and its very detailed preservation (Wappler, 2003). Specimen PE_2000_3-LS fits well within the normal shape scheme of male Eckfeld specimens. HLMD-Me-13500 from Messel also provides genitalia and is very distinct from the Titanomyrma-genitalia described by Wappler (2003). It could be determined as T. gigantea. Ironically, the genital apparatus of T. gigantea evidently is much smaller than the one found in the Eckfeld specimen PE_2000_3-LS. A male genital apparatus of T. simillima is still absent to prove a clear distinction between Eckfeld and Messel beyond their shape differences.

The similarities in the “simillima-morphogroup” are probably due to symplesiomorphic characters in wing venation and shape whereas the contractions of wing veins in T. gigantea may represent a reduction from the original state. This inference can be drawn from evolutionary reduction of the wing venation in ants and the comparison of simillima wing venation to other ants with a generalised wing venation (Fig. 9; Brown & Nutting, 1949; Perfilieva, 2010).

Figure 9 Generalised wing venation patterns in different ant taxa.

Collection numbers starting with CASENT represents the California Academy of Sciences entomological collection based in San Francisco, CA. (A) Odontomachus coquereli J. Roger, 1861, subfamily Ponerinae (CASENT0049797; Photo credit: Erin Prado, available from www.antweb.org). (B) Mystrium rogeri A. Forel, 1899, subfamily Amblyoponinae (CASENT0001083; Photo credit: Cerise Chen, available from www.antweb.org). (C) The male F. simillimum specimen MeI14092. This specimen has been chosen due to its good preservation and because male F. simillimum have the most basal venation patterns with their well pronounced vein rs-m (Photo credit: Sonja Wedmann). (D) Dolichoderus debilis C. Emery, 1894, subfamily Dolichoderinae (CASENT0902952; Photo credit: Will Ericson, available from www.antweb.org).

Diversity within Titanomyrma

The term “species” and higher ranks of taxonomy have of course inherent flaws as classifications are erected that reduce natural complexity and attempt to group relationships by certain criteria. This has been a problem in palaeontology as speciation in the fossil record has been a central discussion since the middle of the 20th century (Sepkoski, 2016). Fossils have lost phylogenetic characters in comparison to their living progenitors so it is often hard to state a clear difference between differently aged specimens. GM reveals that there are informative morphological differences between all the Titanomyrma ants despite their similarity at the first glance.

In many ant lineages, intraspecific cases of bimodal size variation are reported, especially in females, which produced the term “queen-size-dimorphism” (QSD; Heinze & Tsuji, 1995; Wolf & Seppä, 2016b). So far, no hypotheses have been made regarding T. gigantea and T. simillima as polymorphic and conspecific although there is a bimodal size distribution. Causes for QSD are often attributed to alternative strategies of reproduction; e.g. larger queens, the macrogynes, are better suited for founding colonies far off from their birthplace whereas the microgynes settle near their birthplace or are even incorporated into their birth-colony (Heinze & Tsuji, 1995; Rueppell & Heinze, 1999; Wolf & Seppä, 2016b). Male size dimorphism is far less common but when it appears, it may have the same causes as QSD and the two forms can overlap (Fortelius et al., 1987; Elmes, 1991; Heinze & Tsuji, 1995; Wolf & Seppä, 2016a). In the Messel sample, this could mean that there are simply two morphs for each sex belonging to a single species that uses different reproductive strategies. However, the bimodality in size that is also observable within Titanomyrma males is atypical (Rueppell, Heinze & Hölldobler, 1998). Apart from size and wing venation, there is no evidence for the subdivision of T. gigantea and T. simillima into two different species (Lutz, 1986, 1990). A case concerning QSD in ant GM could distinguish macrogynes and microgynes but not after size-relevant factors had been removed (Perfilieva, 2007). As there is clear shape discrimination between T. gigantea and T. simillima that also impacts male wing shape, GM analyses strongly support a two species interpretation.

For the deposits of Eckfeld and Messel, there is a 5 Ma difference between the organisms and theoretically, it would be possible that one of the species from Messel had survived over that time, which would make most specimens from Eckfeld representatives of T. simillima due to their similarities in shape. For some invertebrates, stasis has been reported and speciation events occur in punctuated equilibria over an undisclosed amount of time (Benton & Pearson, 2001). But observations in the Formicidae, for example in Formica rufa C. Linnaeus, 1761 and relatives, show that speciation can take place in a relatively short span lesser than 200 ka (Goropashnaya, Fedorov & Pamilo, 2004). Given that the differences between males of T. simillima and males from Eckfeld in their shape are easy to see in PCA and CVA, classifying Eckfeld specimens as T. simillima is not advisable. A closer association with T. gigantea males has also been disproven because wing shape is significantly different in the Eckfeld males despite ranging in the same size. The significance of these characters in distinguishing a new species can be accepted as in Titanomyrma, the wings are the most informative dividing factor. With PE_1998_17-LS, a second species in Eckfeld is suggested that has to be treated with caution. It stands out in every aspect of morphology but is singular and incomplete. For a final description of Eckfeld species, more distinct characters such as the male genitals in PE_2000_3-LS should also be considered and compared.

Lutz (1986) acknowledged the wing venation differences between F. brodiei and Titanomyrma, but observed affiliations between males of T. simillima and F. brodiei due to size, which cannot be supported when analysing shape patterns. In fact, according to his measurements, the F. brodiei holotype is the same size as Eckfeld and T. gigantea males. F. brodiei and F. mirabile may be closer in shape to the Eckfeld specimens than to T. simillima. Size is a relevant factor in distinguishing the giant ants and those specimens fit very well together. The giant ants from England and Eckfeld should be compared regarding the expression of rs-m and M2. This task could be hard as Lutz (1986, 1990) noted heavy damages and deformation for the holotypes of F. brodiei and F. mirabile. A reinvestigation using the original fossils from the Bracklesham group is advisable to confirm the species status of the Eckfeld specimens and as the two species F. brodiei and F. mirabile may even represent the male and female of a single giant ant species (Lutz, 1986, 1990).

Titanomyrma lubei from the Green River formation is very similar to T. simillima but is not conspecific due to gaster shape (Archibald et al., 2011). No wing venation is recorded from T. lubei and a larger set of specimens beyond the holotype will shed new light on the relationships to the German giant ants. Currently, two new specimens with well-preserved spiracles and anterior parts of the body are being investigated (S.B. Archibald, 2017, personal communication). As long as there is no wing venation for T. lubei at hand, there is no possibility to investigate relationships to the geographically closer F. berryi.

GM for investigating ant relationships in deep time

Reconstructing ant phylogeny based on morphology is difficult, in part due to convergent behavioural traits, such as seed harvesting and specialised predation, or homoplastic morphological adaptations like polymorphic worker castes, so molecular phylogenies are valuable for shedding light on evolutionary relationships within ants (Hölldobler & Wilson, 1990; Moreau et al., 2006; Ward, 2007; Arnan et al., 2012; Ward et al., 2015). While the practice of utilizing shape patterns to reconstruct phylogenetic relationships has been critically reviewed (De Meulemeester et al., 2012), it is becoming increasingly popular (Klingenberg & Gidaszewski, 2010; Adams, Rohlf & Slice, 2013).

Here, many fossil specimens could be included due to the thorough collecting work that has been done in the Messel and Eckfeld fossil localities but in other analyses, far smaller sample sizes are to expect. For bee fossils, GM has already been applied to trace taxonomic affinities and diversity over time, despite the scarcity of available wing specimens (De Meulemeester et al., 2012; Wappler et al., 2012; Dehon et al., 2014, 2017). Singletons like F. brodiei or F. berryi do not offer the same thorough investigation possibilities like the larger number of specimens from Messel and Eckfeld but in shape comparison to the others, affinities for F. brodiei to the Eckfeld-morphogroup or a more stand-alone F. berryi are evidenced by GM so either way, the single fossil is informative.

A properly formed ant wing is not a constant selective factor. This could result in high wing-shape variability over the course of evolutionary history. The most extensive study on patterns of ant wing venation in an evolutionary context was carried out by Perfilieva (2010) but ant wing venation was found to be an unreliable phylogenetic character as any state of reduction occurs in two or more subfamilies simultaneously. Nevertheless, ant wing-venation has been proposed as a valuable morphological character and there are individual variations and trends at the subfamily level (Brown & Nutting, 1949; Klingenberg & Dietz, 2004; Perfilieva, 2010). Our results support the idea that wing venation in ants is an informative character to differentiate groups and that the methods demonstrate potential to make use of fossil ant wings even when preservation is incomplete.

With GM, palaeontology could be further incorporated into formicid evolutionary research, especially to include valuable compression fossils that often only preserve isolated wings. For example, several species of Cretaceous “armaniid” aculeate fossils from Russia and Africa have been proposed as early members of Formicidae (Dlussky, 1975, 1983; Bolton, 2003). However, because these taxa are known only from winged female imprint fossils, it has remained impossible to confidently identify key synapomorphies typically utilised to place ant taxa (Engel & Grimaldi, 2005; LaPolla, Dlussky & Perrichot, 2013). Because many of these armaniid fossils preserve wing venation, it may be possible for GM to show affinities of these enigmatic taxa, which in turn could significantly improve the understanding of early ant history. Studies that incorporate GM could also aid in reinvestigating various interesting fossil ants that better resemble extant ants, such as the large poneromorph queens from the Paleogene of Denmark (Rust & Andersen, 1999). The almost 100 specimens were assigned to the extant poneromorph genus Pachycondyla F. Smith, 1858 because of strong similarities in the head but were revised to the fossil myrmeciine genus Ypresiomyrma, also due to wing venation characters (Rust & Andersen, 1999; Archibald, Cover & Moreau, 2006). A further GM analysis of such specimens could be used to both reinvestigate previous attributions and trace ant evolutionary history in a larger scale.

Position of Titanomyrma in the ant tree of life

The relationships between Formiciinae and other subfamilies within Formicidae are enigmatic, primarily because the giant ants are quite unique. For example, the combination of a generalised wing venation pattern and a reduction of rs-m as in T. gigantea was not observed in Perfilieva’s (2010) study on wing venation but the shape is similar to formicoid ants that possess a reduced M3.

The simillima-morphogroup has the most generalised wing venation pattern in the Formiciinae. Similarities to these giant ants occur for example in Amblyoponinae, Ponerinae, and Dolichoderinae (Fig. 9). Perfilieva (2010) observed a wing venation pattern like in the simillima-morphogroup only in Dolichoderinae and Ponerinae. The differences and similarities to these groups are probably describable and analysable by means of GM. The poneromorph representatives have a much larger relative size of the wing cells 1M, 1-2R, and 1Rs, of which the shape has been analysed here in Titanomyrma. The latter has been associated with these groups for quite some time as the outdated synonym Eoponera suggests (Carpenter, 1929). That synonym has only been revised rather recently by Lutz (1986). Since then, Titanomyrma has been argued to be a basal representative within the formicoid clade or sister taxon to the Formicinae (Lutz, 1986; Wappler, 2003).

Using the wing venation, the symplesiomorphic state of the simillima-morphogroup makes an internal phylogeny of Titanomyrma impossible. The phylogenetic position of Titanomyrma is unresolved and there is no possibility of a phylogeny based on wing venation alone as shown in other studies concerning wing venation (Michez et al., 2009; De Meulemeester et al., 2012; Wappler et al., 2012). However, affinities towards the dolichoderine wing venation pattern and others could be examined using GM (Fig. 9D). The current status of the Formiciinae as sister group to the Formicinae is solely based on the presence of a Furcula in female T. gigantea (Lutz, 1986). Additional useful phylogenetic characters are needed to place the Formiciinae (Grimaldi, Agosti & Carpenter, 1997). GM can be helpful in finding traits. For example, the sexual-shape dimorphism pattern from T. gigantea and T. simillima also occurs in Formicinae, but was not observed in Myrmica (Perfilieva, 2007). Generalised ant wings should be further investigated with robust datasets and it would be desirable to create a dataset among formicid lineages and especially include more plesiomorphic sets of wing venation with a focus on shape to see if the Formiciinae may show more affinities to other ant groups in wing shape.

Palaeoecology

Titanomyrma ants seem to show a surprising diversity on the taxonomic level as well as in the few assumptions that can be made about ecology. T. gigantea dominates in Messel, representing 72% of all giant ant specimens. In Messel, only every fifth specimen is a male whereas in Eckfeld the males are absolutely dominant. In other deposits, the dominance of males or females cannot be assessed from the single wings but because of size, F. brodiei and F. berryi are also possibly males and F. mirabile is probably female. The only published ant from Green River, T. lubei is a female with the size of T. simillima (Archibald et al., 2011).

As the fossils represent a sample accumulated over a certain time, the different ratios of males to females in Messel and Eckfeld are reliable. However, interpreting those differences is highly speculative. Winged ants that accumulated and fossilised in water bodies in some distance to shores have been inferred to originate from mating swarms during nuptial flight (Rust & Andersen, 1999). Ratios of males to females during nuptial flight are difficult to assess and they vary greatly between ant taxa and they can be subjected to intrinsic factors like ant size and extrinsic factors like vegetation, weather, time of the year, or altitude (Nagel & Rettenmeyer, 1973; Franks et al., 1991; Lukasz, 2006; Wolf & Seppä, 2016a). Behavioural explanations for a ratio like this would be that giant ants in Messel relied on “female calling” and in Eckfeld on “male aggregation” syndromes (Wappler, 2003; Boomsma, Baer & Heinze, 2005). Another source of different distribution could be that one of the sexes was the better flyer. A correlation of shape and an increased proportion in the sample is PC 2 in a PCA of SD 9 (see Fig. 2B). A narrower wing means more specimens. The ability of a larger ant to settle and fly farer from the birthplace compared to a smaller ant could explain a higher proportion of Titanomyrma females in the sample. Weight reconstructions in comparison with wing morphology imply that giant ants were poor flyers overall, which favoured falling into the steady water bodies and rapidly being fossilised (Wappler, 2003).

The Eocene gigantism also remains enigmatic in other ecological aspects. Reconstructed temperatures from the fossil localities of the giant ants speak in favour of thermophily (Archibald et al., 2011), but at present, there is not enough evidence to say that the Palaeocene–Eocene Thermal Maximum actually is the cause of gigantism (Verberk & Bilton, 2011; Vermeij, 2016). The giant ants had an Arctic dispersal with the additional finds from North America (Carpenter, 1929; Archibald et al., 2011). Other ecological assumptions concerning Titanomyrma are even more difficult, especially as workers are absent. The abundance of Titanomyrma in Messel is outstanding but their size may prohibit a large colony or nest size although extant “giants” like Dinomyrmex gigas (P.A. Latreille, 1802) do have multi-nest colonies of about 7,000 inhabitants (Lutz, 1986; Pfeiffer & Linsenmair, 2000; Ward, Blaimer & Fisher, 2016).

Conclusion

Geometric morphometrics, applied to fossil Titanomyrma specimens that are at least 50% complete in the section of the wing analysed here, reveals strong differences between groups. To investigate affinities between closely related ants based on their wing venation, several GM analyses are necessary because variation in the dataset is subtle and shape trends like a narrower wing in larger ants have an influence on the overall variation within the dataset. Creating subdatasets aids in finding visualizations and tracing the subtler differences.

Within the global dataset, LDA supports different groupings according to putative sex and species. A significant difference between all wings that have been described as species so far is observable. Moreover, all specimens that have been grouped in species are closer related to each other than to other giant ants and previously undetermined specimens are assignable using GM. The same applies to Titanomyrma sp. from Eckfeld where most specimens can be interpreted as belonging to a single species that is not yet described. Low variation makes sexual dimorphism in wing shape traceable to only some extent. Undetermined specimens from Messel are easily assigned to the species T. gigantea and T. simillima by analysing their wing shape. The results from the PCA are unambiguous and allow species determinations that represent Lutz’s (1986) taxonomical criteria based on the wing venation. Titanomyrma wing venation is generalised. The most significant modifications occur in T. gigantea and this species provides the largest and best-preserved dataset for future analyses. Shape similarities between T. simillima, F. brodiei, F. mirabile, F. berryi, and male Eckfeld Fomiciinae are possibly due to plesiomorphic venation conditions.

For Messel, it is quite remarkable that in a dataset of about 360 wings there are few outliers and that GM works well on two closely related species. Moreover, both species exhibit sexual dimorphism in the same features. Because of the reliability of the landmarks and the significant results, the methods of GM should be applied to gain insights into ant relationships from a morphological point of view. Titanomyrma wing venation should be analysed in context to other subfamilies within the Formicidae to investigate evolutionary patterns in ants and to further illuminate the biology of the giant ants.

Supplemental Information

Supplemental Information 1 A priori Classifiers for Titanomyrma and Formicium specimens.

Click here for additional data file.

Supplemental Information 2 Classifiers for Titanomyrma and Formicium specimens after identification with GM.

Click here for additional data file.

Supplemental Information 3 Previously incorrectly determined, ambiguous, and problematic specimens in the geometric morphometric analyses.

Click here for additional data file.

Supplemental Information 4 P-values, cross-validation results, and Mahalanobis distances from LDA performed with the data out of TPS file Data S2, the classifiers from Table S2, and within MorphoJ file Data S4.

Click here for additional data file.

Supplemental Information 5 TPS file with the Titanomyrma dataset before missing landmarks were estimated.

Click here for additional data file.

Supplemental Information 6 TPS file with the final Titanomyrma dataset.

Click here for additional data file.

Supplemental Information 7 MorphoJ project file with GM analyses of ant fossils with two wings preserved.

Analyses were performed to investigate whether left and right wings of an individual ant are assignable to each other using GM.

Click here for additional data file.

Supplemental Information 8 MorphoJ project file with GM analyses of all available Titanomyrma and Formicium fossils.

For most analyses, the complete dataset (402 wings) was fractioned into subdatasets regarding classifiers and additional data availability (see Tables 1, S1, and S2).

Click here for additional data file.

Supplemental Information 9 R code to import TPS with missing values and export interpolated TPS files based on Schlager (2017, see references).

Click here for additional data file.

Supplemental Information 10 Geometric morphometric analysis exclusively concerning giant ants from Eckfeld.

(A–H) Analyses were performed in SD 12. (A) PCA of all Eckfeld specimens, females appear on the opposite extreme values in PC1 and the other specimens cluster closer together in the centre. There is no pattern of division in any of the PCs. (B) Five larger Eckfeld specimens highlighted in a PCA without females. (C) Loadings of the PCs from the PCA visualized in B. (D) Separating the five larger specimens in a CVA reveals that there is almost no shape difference between the two groups. (E–H) The analyses were performed after a pooled-within-group regression (E) with the treatment of the five larger specimens as a group. (F) A PCA after the attempt for size correction does not distinguish the putative groups. (G) Loadings of the PCs from the PCA visualized in F. (H) Separating the five larger specimens in a CVA after the attempt of size correction reveals that there is almost no shape difference between the two groups.

Click here for additional data file.

This is contribution no. 149 of the series Fossilfundstätte Eckfelder Maar (Mittel-Eozän). We would like to thank Jes Rust (Bonn, DE) for reviewing the earliest versions of this manuscript. The authors would like to thank Herbert Lutz (Mainz, DE) for access to collection material. We greatly appreciate the help from Sonja Wedmann (Messel, DE) and Uta Kiel (Messel, DE) from the Senckenberg Research Institute Frankfurt/Main for access to collection material and preparation of photographies. We would like to thank Thorsten Plogschties (Bonn, DE) for help in preparing vector drawings of Titanomyrma wings. We would like to thank Ksenia Perfilieva (Moscow, RU) and two anonymous reviewers for helpful comments on an earlier draft of the manuscript. Additionally, Julian Katzke is grateful to Jes Rust for steady guidance during this work.

Additional Information and Declarations

Competing Interests

Author Contributions

Data Availability

The authors declare that they have no competing interests.

Julian Katzke conceived and designed the experiments, performed the experiments, analysed the data, wrote the paper, prepared figures and/or tables, reviewed drafts of the paper.

Phillip Barden analysed the data, wrote the paper, reviewed drafts of the paper.

Manuel Dehon analysed the data, wrote the paper, reviewed drafts of the paper.

Denis Michez analysed the data, wrote the paper, reviewed drafts of the paper.

Torsten Wappler conceived and designed the experiments, analysed the data, contributed reagents/materials/analysis tools, wrote the paper, prepared figures and/or tables, reviewed drafts of the paper.

The following information was supplied regarding data availability:

Katzke, Julian; Barden, Phillip; Dehon, Manuel; Michez, Denis; Wappler, Torsten (2017): Katzke et al. Titanomyrma Raw Data. https://doi.org/10.6084/m9.figshare.5405107.

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
