# Peer review of "Giant ants and their shape: revealing relationships in the genus Titanomyrma with geometric morphometrics"

_PeerJ, doi:10.7717/peerj.4242_

## Round 0.1 · original submission · Minor Revisions

All three reviewers concluded that the study is interesting and well-executed. They also provided a number of helpful suggestions that could improve the manuscript and the study more generally. In particular, the Reviewers made a number of analytical suggestions that should be followed up with some more analyses.

·

Basic reporting

1. Clear, unambiguous, professional English language used throughout.

Yes.

2. Intro & background to show context.

Yes.

3. Literature well referenced & relevant.

Yes. Only few items can be added. (See "General comments" №№ 2, 3, 5, 21)

4. Structure conforms to PeerJ standards, discipline norm, or improved for clarity.

Yes.

5. Figures are relevant, high quality, well labelled & described.

In general, yes, but I detected missing data in one figure and ambiguous labels in another one. (See "General comments" №№ 4, 10, 14, 15, 17, 19)

6. Raw data supplied (see PeerJ policy).

Yes, presented completely.

Experimental design

1. Original primary research within Scope of the journal.

Yes.

2. Research question well defined, relevant & meaningful. It is stated how the research fills an identified knowledge gap.

Yes.

3. Rigorous investigation performed to a high technical & ethical standard.

Yes.

4. Methods described with sufficient detail & information to replicate.

In general, but some parts need correction, another ones have to be reformulated. (See "General comments" №№ 7, 8, 11, 13, 14, 18, 19)

Validity of the findings

1. Data is robust, statistically sound, & controlled.

In general, but I suppose, that some of the data can be analyzed additionally for more supported results. (See "General comments" №№ 10, 11, 14, 17, 19)

2. Conclusions are well stated, linked to original research question & limited to supporting results.

i. In general, conclusions are well stated, linked to original research question & limited to supporting results. However, I suppose, that some of the data can be analyzed additionally for more supported results. (See "General comments" № 14, 19)
ii. It would be better if authors provide diagnostic keys for the determination of the investigated fossil species.

Additional comments

Comment № 1-(lines:107-108)-You should not use “genus” for the collective group name (Formicium)
Comment №2-(lines:109-110, 116, 117, 119)-Check taxa names. According to International Code of Zoological Nomenclature (article 51.3): «Use of parentheses around authors' names (and dates) in changed combinations. When a species-group name is combined with a generic name, other than the original one, the name of the author of the species-group name, if cited, is to be enclosed in parentheses (the date, if cited, is to be enclosed within the same parentheses).»For example, Titanomyrma gigantea (Lutz, 1986), Formicium berryi (Carpenter, 1929)
Comment №3-(lines:119, 121)-Check taxon name. Formicium mirabile, not mirabilis (according Bolton, 2017. An Online Catalog of the Ants…)
Comment №4-(lines:121-Authors write: “The single specimen of F. mirabilis is not included in the analyses.” Why? Please, give reasons.
Comment №5-(lines:195-198, Fig. 1A)-Authors write: “classical nomenclature is slightly modified to be shorter and more stringent”.
In my opinion, this modification is not accurate. Following to the Fig. 1A: Cu1 (or R1) – is the name of the longitudinal vein (it means that it is first branch of cubital (or radial) vein). But M1, M2 etc. look like previous names, but their mean different. M1 and M2 are only parts of longitudinal vein (first or second part (fraction) of medial vein), etc. Moreover, some numbers look like indexes, but not systematically applied, and indexes never appear in the text. I recommend to turn back to nomenclature of Brown and Nutting (1949) or to use actual nomenclature for Hymenoptera (see, for example, works of Denis J. Brothers, Hasan H.Basibuyuk, A. P. Rasnitsyn, K. S. Perfilieva, also at site of American Entomological Institute http://www.amentinst.org )
Comment №6-(lines:213)-Put “reduction cross vein 1r” instead of “reduction first radial vein 1r”
Comment №7-(lines:343-344)-Authors write: “…20 PCs (highest dimensionality in MorphoJ), of which PCs1-6 describe more than 5 % variance and PCs1-3 describe more than 10 % variance.” Obviously error. The sum of PCs1-6 can not be less than the PCs1-3.
Comment №8-(lines:357-363)-Use here and in the whole text “left and right side” to name forewings of one specimen. Because a part and counterpart in paleoentomology, are the matching halves of a compression fossil, a fossil-bearing matrix formed in sedimentary deposits. Therefore, it may be a confusion.
Comment №9-(lines:380)-Put “connection between M+Cu and anal veins” instead of “connection between the medial and anal veins”
Comment №10-(lines:388-390, Fig. 2A)-Better for understanding, if objects of observation will be labelled in Fig. 2A
Comment №11-(lines:416)-Obviously error: “male wings being narrower” should be “female wings being narrower”
Comment №12-(lines:447)-Misprint “0”
Comment №13-(lines:450)-centroid size, not “size”
Comment №14-(lines:457-459, Fig. 4, 551-552)-“Five Eckfeld specimens fall in the range of small T. simillima females but are nevertheless assigned as males: PE-1990-582_Mir, PE-1992-258_Mir, PE-1992-506_Mir, PE-459 2000-15_Mir, and PE-2000-18_Mir.” Further explanation of this decision is not supported by the calculations. I think that authors can present a range of wing lengths for these specimens and mark specimens in Fig. 4. Moreover, to support this decision (all Eckfeld specimens are males), it will be better to analyze only Eckfeld sample. Authors analyze Messel sample separately (topic “Shape discrimination between Messel species” Fig. 3), but not Eckfeld sample.
Comment №15-(lines:465)-Fig. 4A, not 5A
Comment №16-(lines:467)-Misprint “0”
Comment №17-(lines:Fig. 4C)-Missing two objects Formicium berryi and F. brodiei
Comment №18-(lines:499)-“The negative loadings of PC1 that separate males of T.simillima…” I’m sure, that authors mean “values” instead of “loadings”. In this case, it is not the same. Loadings in PCA are related with variables (parameters, features) but not with objects. Please, check this terminology in the whole text. I see correct application in the lines 504 - 505.
Comment №19-(lines:517-518, Fig. 7)-“The shape-similarity between other specimens from Eckfeld with PE_1998_17-LS in PC1 and PC2 leads to the assumption that they are conspecific.” Sorry, but I don't see shape similarity in figure 7. Why PE_1994_167_LS absent in Fig. 7? In general, two specimens (PE_1998_17-LS and PE_1994_167_LS) needed to be discussed more clearly.
Comment №20-(lines:537-539)-See the comment № 19
Comment №21-(lines:687-689)-See Perfilieva, K.S.[Variability of quantitative characteristics of wings by the example of some ant species (Hymenoptera, Formicidae)] in Russian // Uspekhi sovremennoi biologii. — 2007. — Vol. 127, no. 2. — P. 147–156.

Reviewer 2 ·

Basic reporting

Authors show an interesting study concerning Geometric Morphometrics of ant’s wings that include fossil specimens with the aim of discriminate sexes and species. In general, the literature references and field background are enough to contextualize this study. The English language should be improve to ensure that the international audience can clearly understand the text. I suggest to use professional and unambiguous language. Some examples where the language could be improved include lines 300, 304, 355, 427-428, 495. The current phrasing makes comprehension difficult.

Experimental design

This study represent an original research concordant with the aims and scope of this journal. The research question is well defined and the information obtained may help to the discrimination of these species. The methods described are conceptually detailed but they lack of some relevant information to estimate the robustness of the analysis.

Validity of the findings

Results found could represent a valuable help in the entomological studies of this taxonomic group. However, the statistical analysis, the results as well as Discussion should be improved.

Additional comments

Material & methods should be improved to estimate the robustness of the study. The results as well as the Discussion should be also improved. I suggest the following changes:
• • Please, review the sample size since it is inconsistent in the abstract and the lines 168, 169, 170. Please consider exchanging the current table 1 by a table that condense the number of individuals of T. gigantea, T. simillima, unclassified specimens, classifier used, sex, type of wing (right or left), origin or source of the wing (Messel, Eckarft, …).
• Summarize the analysis. Several analysis of the 12 subdata set are reiterative.
• Include technical information of the pictures (camera, focal distance, so on) or drawing, to estimate possible optical distortion.
• Include the repeatability of digitalization and indicate the number of persons that made the digitalization, which will allow estimate the precision in the digitalization and the role of technical causes on the differences found.
• Please, summarize the idea between the lines 269-275.
• The table 2 evidences variations in the patterns of wing venation in different species and samples. Please, explain how this situation affect your analysis considering the missing landmarks.
• Include the statistical significance of the difference of Procrustes/Euclidian distances provided by the multivariate analysis.
• Please consider exchanging the linear regression analysis by multivariate regression analysis to test the allometric effects.
• Lines 340-363: Please consider include in the MS, figure(s) or table(s) that allow to follow your description in this section. Indicate the percentage of variance explained by the number of PCs included in the text and the P-values.
• Lines 373, 385: Cite figure(s).
• Lines 388-389, 392-393, 399-401: Please, shown the data in the figure.
• Lines 407-410: These results suggest allometric effect. Please, explain why you did not test the hypothesis of common allometric slopes and make correction for size after testing. Include the statistical significance.
• Lines 606-608: In GM, the mixture of data from different persons that makes the digitalization, introduce biased measures. Instead, it is recommended to publish the pictures that may be used by different persons.
• Lines 634 – 636: In this case, it is recommended to compare statistically the magnitude of sexual shape dimorphism.
• Figures 2, 3(C,D), 4, 6, 7(A,B): Please, include the percentage of explained variance in each axis or in the legend of each figure.
• Please consider exchanging the current figure 8 by information included in the text.
• Please consider delete the figures 9 and 10.
• Focus your discussion to supporting results.

Annotated reviews are not available for download in order to protect the identity of reviewers who chose to remain anonymous.

Reviewer 3 ·

Basic reporting

Although English is not my native language, I believe the manuscript is very well written, with an enormous amount of details that are usually only described in reviews or textbooks. I think this detailed description very important. Literature is well used and the article structure is ok.The results are very interesting and important to the field, especially for ants. The applied methods are correct, and the results support the discussion and the most of the conclusions.

Experimental design

I have two significant concerns regarding data acquisition that may compromise the results and, therefore, a large part of the results.
The first concern is related to the use of the two wing drawings of holotypes. It is well known that Geometric Morphometric techniques are very sensitive to detect differences in shapes, even in small scale. The use of drawings introduces a new source of variation other than the operator plotting the landmarks, that are the small distortions that a drawing may contain that are not in the original piece that was used as a model. That said, I would not recommend the use of these specimens in the analysis. If authors decide to maintain the drawings in the analysis, the results must be interpreted cautiously, due to the reasons exposed above.

My second concern is related to the inclusion, in the analysis, of the wings with missing landmarks, using a statistical function to estimate the relative position of these landmarks. Again, it introduces a non-natural variation that may influence the results, minimizing the natural variation. I understand that, for paleontological studies, the introduction of this variation is acceptable, due to the rarity of the samples and the different preservation status of them. However, again, these results should be interpreted cautiously.

Validity of the findings

see comment above.

Additional comments

no comments

---

## Round 0.2 · accepted · Accept

The reviewers had made a large number of minor suggestions, and the authors have done an excellent job addressing them. I recommend that the manuscript be published as is.